# Antibody Focusing to Conserved Sites of Vulnerability: The Immunological Pathways for ‘Universal’ Influenza Vaccines

**DOI:** 10.3390/vaccines9020125

**Published:** 2021-02-05

**Authors:** Maya Sangesland, Daniel Lingwood

**Affiliations:** The Ragon Institute of MGH, MIT, and Harvard, 400 Technology Square, Cambridge, MA 02139, USA; msangesland@mgh.harvard.edu

**Keywords:** influenza virus, antibody response, universal vaccine, immunodominance, broadly neutralizing antibodies, B cell immunology

## Abstract

Influenza virus remains a serious public health burden due to ongoing viral evolution. Vaccination remains the best measure of prophylaxis, yet current seasonal vaccines elicit strain-specific neutralizing responses that favor the hypervariable epitopes on the virus. This necessitates yearly reformulations of seasonal vaccines, which can be limited in efficacy and also shortchange pandemic preparedness. Universal vaccine development aims to overcome these deficits by redirecting antibody responses to functionally conserved sites of viral vulnerability to enable broad coverage. However, this is challenging as such antibodies are largely immunologically silent, both following vaccination and infection. Defining and then overcoming the immunological basis for such subdominant or ‘immuno-recessive’ antibody targeting has thus become an important aspect of universal vaccine development. This, coupled with structure-guided immunogen design, has led to proof-of-concept that it is possible to rationally refocus humoral immunity upon normally ‘unseen’ broadly neutralizing antibody targets on influenza virus.

## 1. Introduction—The Need for Universal Influenza Vaccines

Viral pathogens such as smallpox and measles have been completely or nearly eradicated on a global scale owing to the remarkable success of vaccines; however, influenza virus continues to remain a critical public health issue, as repeated exposure through infection or yearly vaccination has yet to yield long-lasting and durable protection [1,2,3,4,5]. This inability to generate broadly protective herd immunity imposes a significant burden to healthcare systems, where influenza virus is responsible for roughly 3–5 million cases of infection globally with up to 650,000 annual deaths [6,7].

Influenza viruses belong to the family Orthomyxoviridae, which are enveloped viruses containing segmented RNA genomes that can infect across avian and mammalian species, including humans [8] (Figure 1A). Of the four influenza genera (A, B, C, and D), influenza A and B viruses are responsible for infections in humans. However, influenza A viruses (IAVs) tend to receive more concern, as they evolve faster, incur higher rates of morbidity and mortality, and harbor potential for future pandemic outbreaks [9,10]. IAVs are segregated into subtypes based on the antigenic and phylogenetic characteristics of the two surface glycoproteins hemagglutinin (HA) and neuraminidase (NA). The CDC currently designates 18 hemagglutinin (H1–H18) and 11 neuraminidase (N1–N11) subtypes that combine to form the viral subtype (e.g., H1N1), with the potential for 198 IAV subtype combinations [11]. HA can be further subdivided into Group 1 or Group 2 based on further antigenic variation (Figure 1B). Of the influenza subtypes, H1N1 and H3N2 strains routinely co-circulate in the human population, and, along with influenza B, are responsible for yearly seasonal epidemics [7,12]. Current seasonal vaccines consist of a trivalent or quadrivalent formulation which includes two influenza A strains (H1N1 and H3N2) and one or two strains from the influenza B lineages (Yamagata and Victoria) (Figure 1B). Annual vaccination remains the best countermeasure against disease, yet efficacy can range from 10% to 60% in a given year and offers little protection from novel pandemic strains [13,14]. Variability in seasonal vaccine efficacy is, in large part, linked to the highly mutable nature of the virus, which encodes an error-prone RNA polymerase, resulting in the accumulation of mutations in the two major surface antigens, hemagglutinin and neuraminidase, through a process known as antigenic drift [15,16]. By introducing both strain and subtype variability, antigenic drift complicates traditional vaccine approaches by supporting viral escape from pre-existing immunity [15,16,17,18,19]. Vaccines are thus reformulated yearly based on surveillance measures and predictions published by the World Health Organization [20], yet as breadth of protection from seasonal vaccinations is often narrow and largely strain-specific, efficacy suffers in years when formulations are discordant with circulating strains [13,21,22].

Pandemic strains can emerge through antigenic shift, a process by which two different IAV strains of zoonotic origin (including avian and swine species) combine to form a new subtype that is infectious in humans. This can be extremely dangerous as newly emerging subtypes are often antigenically novel with no pre-existing immunity [23,24]. Indeed, in the last ~100 years, four major IAV pandemics have occurred: the H1N1 Spanish influenza (1918), the H2N2 Asian influenza (1957), the H3N2 Hong Kong influenza (1968), and most recently, the H1N1 swine influenza in 2009. In each of the above examples, the pandemic arose either directly from an avian host into humans (1918 pandemic) or through reassortment events between avian–human viruses (1957 and 1968 pandemics) or between swine–avian–human viruses (2009 pandemic) [25,26,27,28]. Ongoing pandemic concerns center on the emergence of novel H5N1 and H7N9 viruses originating from avian species, which have already caused isolated spillover events and disease outbreaks in humans [29,30].

Given the limitations of seasonal vaccines and the continual threat of pandemic IAVs, the development of universal influenza vaccination strategies remain a priority [4,31,32,33,34,35,36]. A central goal is to develop vaccines that will provide protection not only against current circulating strains, but also across subtype diversity, encompassing any future strains arising from antigenic shift or drift [16,18,19]. Ideally, such a vaccine would provide ≥75% protection against both Group 1 and Group 2 IAVs with durability lasting at least 1 year [31,36]. The development of universal vaccines has been further guided by the discovery and ever-growing list of influenza broadly neutralizing and broadly protective antibodies (bnAbs) that target conserved sites of viral vulnerability, particularly those that interfere with HA function [37,38,39,40,41,42,43,44,45,46,47,48,49,50,51,52,53,54,55,56,57,58,59,60,61,62,63,64,65,66,67,68,69]. Broad-spectrum antibodies also engage conserved targets on other viral surface proteins, including neuraminidase and M2 [70,71,72,73,74,75,76,77]. In this review, we outline and discuss the immunological basis and feasibility of rationally engineering humoral immunity that is refocused upon such target epitopes.

## 2. Subverting Natural Immunodominance Hierarchies: An Immunological Challenge for Universal Influenza Vaccines

A unifying feature of influenza broadly neutralizing/broadly protective antibody responses is their immunological subdominance, a major stumbling block for universal vaccine development. Defined as the tendency of the immune system to respond to complex antigens in a hierarchical manner, the immunodominance patterns enforced following influenza infection and vaccination prioritize the expansion of non-neutralizing antibodies against ‘off-target’ hypervariable features at the expense of ‘on-target’ responses engaging functionally conserved epitopes [1,78,79,80,81,82,83] (Figure 2). Structure-based influenza immunogens applied within transgenic mouse systems bearing user-defined B cell receptor (BCR) repertoires, along with similar approaches using HIV antigens, have enabled experimental manipulation of these parameters [84,85,86,87,88,89,90,91,92]. This work indicates that the immunogenicity of a given epitope is a function of the frequency and affinity of the on-target versus off-target BCRs present in the antigen naïve germline repertoire. Subdominant antibody responses thus arise when low frequency and/or low affinity on-target BCRs are unable to compete for expansion within B cell germinal centers (GCs), allowing off-target (‘immuno-distracted’) B cells to then dominate the GC reaction [84,85,86,87,88,89,90,91]. Such off-target anti-influenza BCRs proceed to overshadow antibody affinity maturation, the downstream serum antibody response, and the development of B cell memory [85,86]. T cell help, which is a limiting factor for GC reactions and the development of B cell memory, also appears to be skewed to promote the expansion of off-target anti-influenza B cells [83,93].

While the physiochemical basis of low-frequency and/or low-affinity on-target BCRs remains unclear, the immuno-distractive properties of influenza viral surface antigens appear to be universally maintained, given that immunodominance hierarchies of influenza antigens are recapitulated in jawless vertebrates which utilize a convergent but non-immunoglobulin form of antibody-like antigen engagement [78,79]. Hence at the immunological level, a significant challenge of universal influenza vaccine development centers on the ability to overcome these hardwired rules of immune distraction, so as to selectively promote the expansion of normally immunologically silent B cells with specificity for conserved sites of viral vulnerability.

## 3. Functionally Conserved Antibody Targets on Influenza Hemagglutinin

The viral spike glycoprotein hemagglutinin currently serves as the primary seasonal influenza vaccine antigen, but also bears a number of conserved features which have been proposed as broadly protective ‘universal’ vaccine targets [1,4,31,34,35,94,95]. HA is a homotrimer consisting of a highly variable globular head domain (HA1 subunit) and a more conserved membrane proximal stalk region (HA2 subunit) [94,96,97,98]. Structurally resolved sites of vulnerability are described in detail below and include targets on both the head and stalk domains (Figure 3). Notably, antibodies engaging these sites are immune-recessive and will necessitate engineered refocusing strategies for reproducible elicitation through vaccination.

### 3.1. HA Head Epitopes

While largely hypervariable, the head domain contains a functionally conserved receptor binding site (RBS) pocket that engages cell surface sialyl oligosaccharide, the primary receptor for influenza virus [97,99]. Structurally, the RBS is composed of four loops that form the outer ridges (the 130 loop, the 150 loop, the 190 helix, and 220 loop) and four highly conserved residues within the base of the pocket [96]. Neutralizing antibodies detected by hemagglutination inhibition (HAI) block access to this site and are most typically elicited by seasonal vaccines, where they serve as a primary correlate of protection [13,100,101,102]. Such neutralizing antibodies are largely strain-specific due to interactions with the surrounding hypervariable features on the HA head domain [16,18,19,54]. However, broadly neutralizing activity can be conferred when peripheral non-RBS contacts are minimized [54,55,56,57,58,59,60,61,103]. Both homosubtypic and heterosubtypic neutralization activities have been reported for RBS bnAbs [55,56,57,59,60,61,62,63,64,65], and functionally convergent antibody paratopes underscore some of these responses, including long CDRH3 loops that mimic sialyl oligosaccharide [54,57,61,62,63,64,104,105]. Broadly neutralizing activity can also be conferred by engaging conserved HAI-negative targets on the HA head region, including the vestigial esterase domain [106,107] and lateral patch epitope [108] (Figure 3).

Recently, a conserved site of vulnerability has been identified at the interface of two HA protomers on the head domain [109,110,111]. This interface epitope is hidden within the trimeric prefusion conformation but may become transiently accessible through dynamic conformational changes associated with ‘breathing’, which has been measured for HA trimers at a neutral pH [112]. During infection, the virus is internalized within acidic endosomes and these the dynamic fluctuations give way to pH-triggered rearrangements in HA2, leading to fusion of the viral and cellular membranes [112]. Passive transfer studies in mice have demonstrated that these antibodies confer broad protection through antibody-dependent cellular cytotoxicity (ADCC) and complement-dependent cytotoxicity (CDC) [109,110,111]. Such antibody Fc functionality also underscores the protective activity of other broad-spectrum head antibodies [113,114], including those engaging the vestigial esterase domain [106,107]. By contrast, antibodies with RBS-blocking activity tend to induce less robust ADCC responses [115,116,117,118]. Nevertheless, antibody neutralization and induction of ADCC are not mutually exclusive and can operate synergistically [106,119,120]. Such synergy may also be balanced by antibody-dependent enhancement risks, such as enhanced infection of monocytes and myeloid dendritic cells as reported for anti-HA antibodies from macaques [121].

### 3.2. HA Stalk Epitopes

The HA stalk is relatively conserved as compared with the head domain and, as such, readily supports broad heterosubtypic neutralizing activities [1,10,54,122]. Stalk bnAbs target a highly conserved hydrophobic groove within the HA1/HA2 interface, which is functionally preserved due to its role in virus–host cell membrane fusion [10,123] (Figure 3). Humans can generate low-titer antibody responses against the HA stalk, which has supplied an ever-growing list of stalk bnAbs encompassing Group 1 IAVs [37,38,39,40,41,42,43,53], Group 2 IAVs [44,45,46,47], Group 1 and Group 2 IAVs [48,50,51,67,68], as well as across both influenza A and B viruses [49,52]. Structural characterization of these antibodies has indicated that pan-Group 1 neutralizers are unable to contend with an N-glycosylation site at HA1 Asn38 which is present on Group 2 HAs [37,38,39,40,41,42,43]. Pan-Group 2 neutralizers avoid this clash by targeting a spatially distinct epitope that is closer to the base of the stalk; however, this comes at the expense of Group 1 neutralization [44,45,46,47]. Importantly, several pan-IAV bnAbs are able to accommodate the Asn38 glycan on the stalk, bearing footprints similar to Group 1 bnAbs but with elevated neutralization breadth [48,49,50,51,52,53,66,67,68,69].

Stalk bnAbs have been operationally defined as inhibitors of viral membrane fusion, a neutralization activity that, while broader, has lower potency as compared with antibodies that block the RBS [124,125]. High titers of prophylactic stalk bnAbs can mediate protection from diverse influenza viral challenges within immunodeficient NOD.SCID.*Il2rg*^−/−^ (NSG) mice, through neutralization alone [126]. However, at lower bnAb titers, passive transfer experiments in mice have revealed that broad protection is also underscored by ADCC [48,113,117,127], which can be enhanced by strengthening Fc functionality [114]. ADCC activity also appears to be enhanced for bnAbs relative to non-bnAb antibodies targeting the HA stalk [116]. More recently, it has been reported that stalk bnAbs can inhibit neuraminidase activity through steric occlusion of NA when adjacent to HA, effectively restricting viral egress [128,129].

## 4. Alternative Universal Vaccine Targets

### 4.1. Neuraminidase

Neuraminidase (NA) is the second most abundant surface glycoprotein and enables viral egress by catalyzing the cleavage of sialyl oligosaccharide from host cell surfaces [130,131] and can aid in early infection in the airway epithelium by removing virus from natural defense proteins such as mucins [132]. Although HA has historically been the primary measure of seasonal vaccine efficacy as well as the focus of more recent universal vaccine approaches, there is increasing evidence that humoral immunity against conserved targets on NA confers broad protection in animal studies [70,71,72,73,74,75,76,77]. Such targeting is also underscored by lower rates of antigenic drift and shift, compared to HA [133,134,135,136], providing an approachable target for universal vaccine development. In humans, antibody responses against NA have long been noted as correlates of protective immunity [137,138,139,140,141,142]. Protection appears to be mediated by direct inhibition of virus budding and egress of viral progeny from infected cells [122], and/or through Fc receptor-mediated activity via ADCC, which has been assessed in vitro [75,143] and by in vivo passive transfer experiments in mice [74,113]. More recently, human broadly neutralizing NA antibodies to the enzyme active sites have been reported to provide coverage across Group 1 and Group 2 IAVs as well as IBVs [75,76].

### 4.2. Matrix 2 Ectodomain 

The matrix 2 (M2) protein of IAV is a transmembrane ion channel involved in viral uncoating during entry [144,145,146]. Notably, M2 contains a 23 amino acid surface exposed ectodomain (M2e) that is highly conserved across IAVs, and has been proposed as a target for universal influenza vaccines [147,148]. Antibodies against M2 are protective [149,150]; however, M2e is also weakly immunogenic [151,152,153]. Generally, M2e antibodies are non-neutralizing, but have been shown to mediate protection through a number of Fc effector functions: NK cell activation, antibody dependent cell-mediated cytotoxicity (ADCC), complement dependent cytotoxicity (CDC), and antibody dependent cell-mediated phagocytosis (ADCP) [154].

## 5. Refocusing Humoral Immunity Upon Conserved Sites of Vulnerability—Proof of Concept

The presence of low-titer broadly neutralizing and/or broadly protective antibodies within human immune sera demonstrates that the basic capacity to trigger and expand these immunoglobulins exists. However, antibody responses to these bnAb targets continue to remain at low and unprotective titers. Computational modeling and mechanistic studies in transgenic mice have indicated that the basis for such immunological subdominance is, in part, due to the low affinity and/or low frequency of on-target germline bnAb precursors, which are unable to compete for selection in the germinal center, resulting in lower contribution to B cell memory and affinity-matured serum antibody responses [85,86,92]. In order to focus the antibody response on these subdominant bnAb targets, universal vaccine strategies will need to: (1) preferentially activate low frequency on-target germline BCRs; and/or (2) preferentially recall low-frequency on-target B cell memory. Such immune-focusing strategies have, in large part, been advanced for HA-based immunogens and will be the focus of this section.

### 5.1. HA Stalk-Based Immunogens

HA stalk-focusing vaccine concepts have received much attention due to the relative conservation and presentation of bnAb epitopes that confer broad heterosubtypic neutralization and protection [1,4,31,34,35]. At the level of B cell memory, a major factor shaping the human antibody response to influenza viral antigens is pre-existing immunity, imprinted by prior infection and/or vaccination [82,155,156]. While such memory is skewed for strain-specific reactivity, resulting in potentially confounding antigenic sin effects, studies in animals have demonstrated the capacity to selectively recall broadly protective B cell memory against conserved stalk epitopes by sequential exposure to chimeric strain-variant HA immunogens (cHA) [157,158,159,160] (Figure 4A). In this strategy, priming and expansion of stalk responses occur through sequential vaccination of cHAs displaying antigenically distinct head domains (e.g., H9, H6, H5) to which humans are immunologically naïve, while containing the same conserved stalk domain from a single HA strain, typically from seasonal H1, H3, or influenza B strains. The immunological premise of this strategy thus relies on the fact that B cell memory against conserved features, which predominate on the HA stalk, will be boosted upon sequential exposure [161,162,163]. As such, these regimens have been developed for antibody focusing to the stalk of Group 1, Group 2, and influenza B HAs [157,158,159,160]. Notably, Group 1 chimeric HA stalk-focusing immunogens were more recently deployed in a Phase 1 clinical trial (NCT03300050). The data released indicate that prime-boosting with chimeric HA vaccines elicited selective boosting of Group 1 stalk antibodies that were maintained for up to 18 months after immunization [164].

Similarly, sequential immunization of humans with HAs from avian H5N1 or H7N9 strains to which the population is normally antigen-naïve, resulted in memory recall of cross-reactive and broadly neutralizing stalk antibodies encompassing both Group 1 [42,43,165] and Group 1/Group 2 IAVs [68,166]. The memory recall of stalk bnAbs can also occur during natural exposure, as evidenced following infection with the highly divergent 2009 pandemic H1N1 virus [40]. Notably, however, pre-existing immunity is often complex and can vary between individuals [82,155,156]. Thus, ensuring the uniformity of broadly protective memory recall will continue to be an important factor moving forward.

In addition to selective expansion of on-target B cell memory, structure-based vaccine design has been applied to engineer stalk-only immunogens, allowing for more efficient engagement of stalk-specific B cells, both at germline and during post-immune memory stages, by eliminating the immune-distracting components of the hypervariable HA head domain [85,86,92,167,168,169,170,171,172,173] (Figure 4B). In animal studies, stalk immunogens have been shown to elicit heterosubtypic humoral immunity that is broadly protective against unmatched Group 1 and Group 2 IAV challenges, including against avian H5N1 and H7N9 strains [86,167,168,169,170]. For most of these immunogens, structure-guided design was applied to array the trimeric stalk-only domains on a self-associating nanoparticle display [168,169,171], an antibody-titer enhancing principle that increases antigen immunogenicity through a combination of elevated BCR crosslinking, increased deposition on antigen-presenting follicular dendritic cells, and enhanced recruitment of B cell clones to GCs [174,175]. Enhanced germline activation through a repetitive antigen array may be particularly important for eliciting Pan-Group 1 and Pan-Group 1/Group 2 immunity in humans, as these stalk bnAbs tend to arise from low-affinity germline BCRs [85,86,169,176,177]. Moreover, structure-guided nanoparticle displays have also been found to more optimally present stalk bnAb epitopes, as compared to their geometric orientation in the native virus [85,92]. Notably, the Group 1 stalk-only nanoparticle H1ssF [86,168] and the Group 2 stalk-only nanoparticle H10ssF are currently under evaluation in Phase 1 clinical trials (NCT03814720 and NCT04579250, respectively).

Manipulating and lengthening the GC reaction time through sustained antigen priming may also facilitate the exploration of the antigenic space and immune discovery of stalk bnAb targets. While formally shown to enhance antibody responses to immunorecessive vaccine targets on HIV immunogens [178,179,180], sustained antigen priming through gene-based expression of transmembrane HA followed by protein boosting was amongst the first demonstrations of vaccine elicitation of stalk bnAbs [181,182]. Structure-guided stalk immunogens [85,86,92,167,168,169,170,171,172,173] or head domain shielded trimers [183], applied in the context of sustained antigen release, may further enhance bnAb elicitation.

### 5.2. HA Head-Based Immunogens

Antibody focusing through selective recall of pre-existing but otherwise immunologically subdominant on-target B cell memory has also been proposed for invariant targets on the HA head domain, namely to the conserved RBS [184,185,186,187]. However, epitope scaffolding in this context is challenging, as the RBS is a complex target, assembled by amino acids that are segregated in sequence space but adjoining within the conformational structure of HA [97]. Nevertheless, structure-guided immunogen design has succeeded in transplanting RBS sites onto appropriately folded heterosubtypic HA recipients [184], potentiating the development of sequential immunization regimens that trigger and expand B cell memory against this bnAb target. In a related approach, enhanced cross-reactivity can also be achieved by sequential immunization with mosaic hemagglutinin trimers, where the immunodominant antigenic sites on the head are replaced with sequences from exotic avian HAs. Sequential immunization with these constructs can elicit antibodies to the stalk, similar to cHA regimens, while simultaneously generating antibody responses to the RBS, as measured by HAI [185,186,187].

Alternatively, broadly neutralizing responses to conserved sites on the HA head domain can be enhanced by the development of nanoparticle immunogens displaying mosaic RBDs, where each monomer is derived from a heterologous H1N1 strain [188] (Figure 4C). The authors propose that presenting strain-variant RBDs on the same molecule, with inter-HA space being roughly equivalent to the spacing required to engage the two arms of the bivalent BCR, would allow for increased BCR crosslinking, thus preferentially activating and expanding B cells with cross-reactive potential, particularly those with specificity for conserved epitopes on the HA head. In support of this, the mosaic RBD nanoparticles elicited broadly neutralizing H1N1 antibody responses targeting a conserved patch proximal and opposite to the RBS [188]. This antibody class, which occurs in humans, was not elicited in the mouse system when structurally equivalent nanoparticles displaying homotypic RBD-monomers were used [188].

The addition of N-glycans can also modify antigenic sites on the HA head and alter recognition by antibodies [189,190,191]. This has been rationally applied to trimer immunogens, where the addition of an increasing number of select glycans can effectively refocus the responding murine B cell repertoire on the broadly protective but otherwise immunologically subdominant interface epitope [109] (Figure 4D). Hyperglycosylation of the HA head domain has also been applied as a means to silence immuno-distractive epitopes and enhance the immunogenicity of subdominant antibody responses against the stalk domain [192], thus indicating that N-glycan addition is a general tool that can be applied to restructure the antibody immunodominance hierarchies elicited by influenza HA.

## 6. Pathway Amplification as an Immunological Principle for Reproducible Vaccine Expansion of On-Target B cells

The human antibody repertoire is highly diverse, containing ~10^12^ possible unique BCRs, and facilitates the accommodation of essentially any antigen, a cornerstone of adaptive immunity [193,194]. However, paradoxically, this same diversity also aids in maintaining the immunological subdominance of bnAbs. For each responding BCR, antigen complementarity is provided by the antibody paratope, which is formed from the CDR1 and CDR2 loops encoded by antibody variable (V) genes, and the hypervariable CDR3 loops, which are assembled de novo by stochastic N-junctional diversification of antibody D and J segments that are unique to each B cell clone [194,195,196]. Repertoire diversity is concentrated within the centrally positioned antibody heavy chain CDR3 (CDRH3) and serves as the principal determinant of antigen specificity [193,194,195,196,197,198,199,200,201]. However, as immunological subdominance is a product of low frequencies of on-target germline precursors [85,86,92], it would appear that antibody targeting by the hypervariable CDRH3 reproducibly fails to engage these conserved bnAb targets. Universal vaccine concepts are thus burdened with eliciting high titer polyclonal antibody responses to conserved sites of vulnerability by means of a baseline targeting system that engages epitopes in a stochastic manner, while concomitantly disfavoring complementarity to the desired targets.

Despite this, the human immunoglobulin repertoire also generates genetically reproducible or ‘public’ responses against immunologically subdominant sites of vulnerability on a number of viruses, including influenza virus, HIV, hepatitis C virus, hepatitis B virus, yellow fever virus, and SARS-CoV-2 [42,68,176,202,203,204,205,206,207,208,209,210,211,212,213,214,215,216,217,218]. These responses are characterized by antigen affinity that is endowed by the antibody V_H_-region-encoded CDRs, resulting in germline-reproducible engagement solutions. Thus, a potential strategy could be to harness this reproducible substrate for epitope complementarity and ‘pathway amplify’ bnAb responses using rationally designed germline-triggering vaccines.

In the context of influenza virus, evidence for such germline endowment arises from the observation that Group 1 neutralizing HA stalk bnAbs are stereotyped for usage of the V_H_-gene IGHV1-69 in humans [37,38,39,40,42,43,165,202,205,206]. Notably, we demonstrated that IGHV1-69 germline BCRs naturally engage the HA stalk bnAb epitope in Group 1 IAVs via the V_H_-encoded CDRH2 loop, providing a basis for V_H_ bias and a potential foundation for reproducible pathway amplification [176,177,219]. Using a transgenic mouse system which mimics human-like CDRH3 diversity but constrains antibody responses to single user-defined V_H_ segments, we further demonstrated that such germline-endowed complementarity can indeed provide a substrate for pathway expansion of bnAbs in response to vaccination [85,86]. Here, we noted that IGHV1-69 specifically endowed the antibody repertoire with a reproducible on-target germline BCR substrate that was then selectively activated and amplified using the structure-guided stalk-nanoparticle H1ssF, resulting in the elicitation of high serum titers of Group 1 bnAbs that were protective against heterosubtypic challenge [85,86] (Figure 5). Notably, pathway amplification of bnAbs did not occur in transgenic animals constrained to a control human V_H_-sequence, demonstrating the gene-endowed nature of this human bnAb response [86] (Figure 5). Importantly, when IGHV1-69 usage was genetically diluted to match the frequency measured in humans, the V_H_-endowed reproducibility was still sufficient to pathway amplify the bnAb response using H1ssF [85,86], suggesting that bnAb elicitation may indeed be possible in the on-going Phase 1 clinical trial for this immunogen (NCT03814720). Interestingly, public antibody targeting through V_H_-endowed affinity is emerging as a general phenomenon for diverse microbial targets [216], suggesting other settings from which genetically pathway-amplifiable antibody responses may be elicited.

## 7. Conclusions

While immunologically recessive, the discovery of human influenza bnAbs has demonstrated a baseline capacity to elicit broadly protective humoral immunity against influenza virus. However, both influenza virus and influenza viral antigens impose complex patterns of immuno-distraction that serve to prevent efficient bnAb output. The use of structure-guided immunogens, along with a more nuanced understanding of the molecular and cellular basis for immunological subdominance has catalyzed the development of proof-of-concept antibody refocusing strategies which have elicited varying degrees of re-engineered immunity and enhanced protective breadth in preclinical studies. Such rationally designed vaccine concepts are now ‘graduating’ to early clinical studies, meaning that while the problem is still far from solved, it may yet be solvable.

## Figures and Tables

**Figure 1 vaccines-09-00125-f001:**
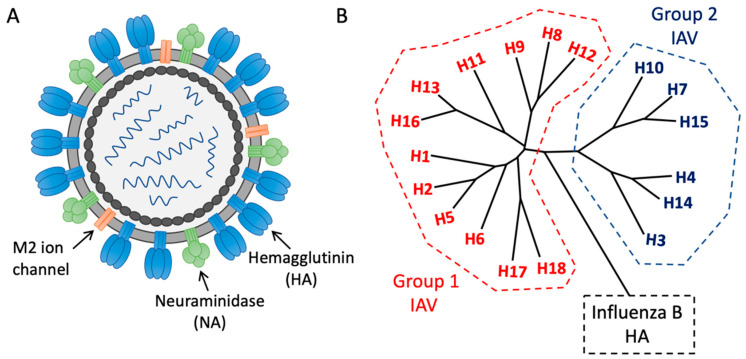
Structure and diversity of influenza virus. (**A**) Influenza is an enveloped virus containing a segmented RNA genome. The surface glycoproteins hemagglutinin (HA) and neuraminidase (NA), along with the M2 ion channel, which spans the viral envelope, serve as potential universal vaccine targets. (**B**) Influenza A (IAV) hemagglutinin is subdivided into Group 1 and Group 2 based on antigenic variability.

**Figure 2 vaccines-09-00125-f002:**
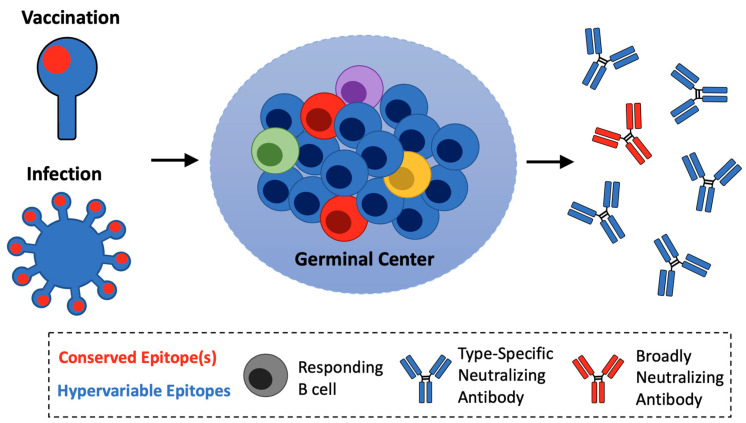
Immunological subdominance of broadly neutralizing responses. Influenza infection and vaccination preferentially elicits antibodies to hypervariable ‘off-target’ epitopes (blue) at the expense of ’on-target’ conserved sites of vulnerability (red). ‘On-target’ broadly neutralizing and broadly protective antibodies (bnAb) precursors fail to compete for selection in the B cell germinal center.

**Figure 3 vaccines-09-00125-f003:**
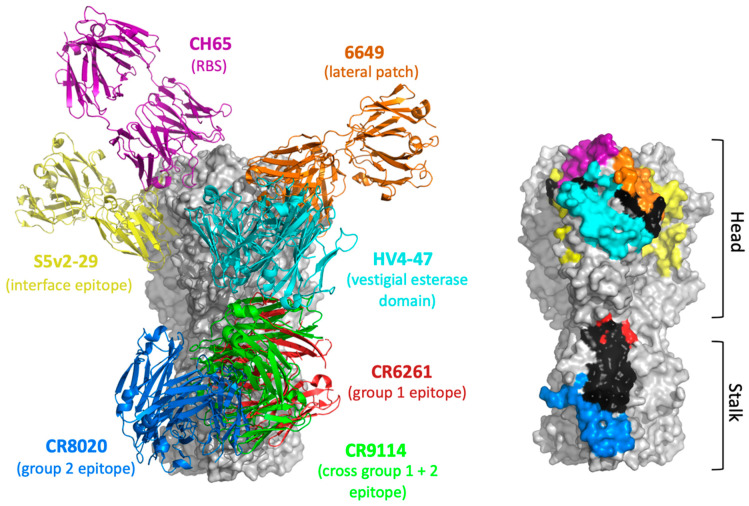
Broadly neutralizing epitopes on hemagglutinin (HA). Shown are (left) monoclonal antibodies (mAbs) and (right) their binding sites, defined by residues with atoms within 5A of their respective mAbs, modeled onto HA PBD 1RU7. Coloring between the modeled mAbs and binding sites are consistent with CR6261 (red, Group 1 epitope), CR9114 (green, Cross-Group 1 + 2 epitope), CR8020 (blue, Group 2 epitope), S5v2-29 (yellow, interface epitope), CH65 (purple, receptor binding site (RBS)), 6649 (orange, lateral patch), and HV5-47 (cyan, vestigial esterase domain). Binding sites are shown on a single protomer for all mAbs except for S5v2-29, which is shown on two protomers, since the site is hidden in this view of the single protomer. Residues which were composed of two or more binding sites are shown in black, while those not within any binding sites remain in gray. Head and stalk domains are labeled accordingly.

**Figure 4 vaccines-09-00125-f004:**
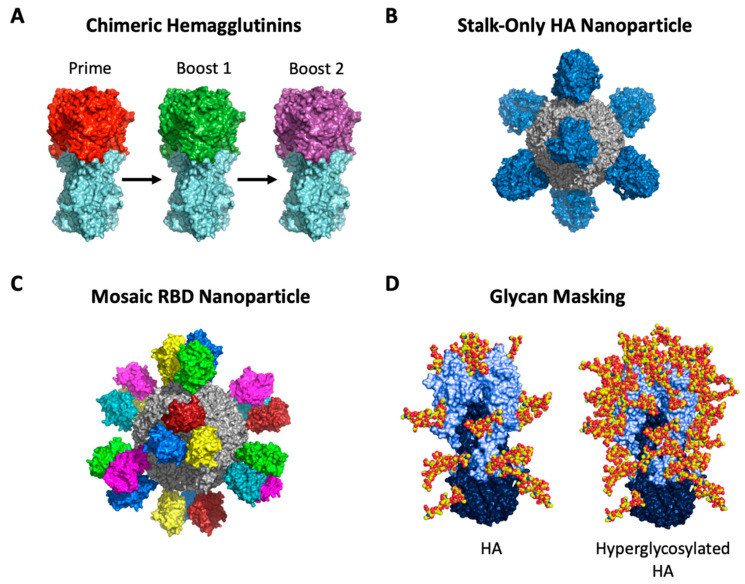
Select immune refocusing strategies to elicit broad cross-reactive HA stalk- and head-targeting antibodies. Strategies include (**A**) sequential immunization with chimeric HA (cHA) immunogens, (**B**) stalk-only HA nanoparticle immunogens, (**C**) mosaic RBD nanoparticles, and (**D**) glycan-masking.

**Figure 5 vaccines-09-00125-f005:**
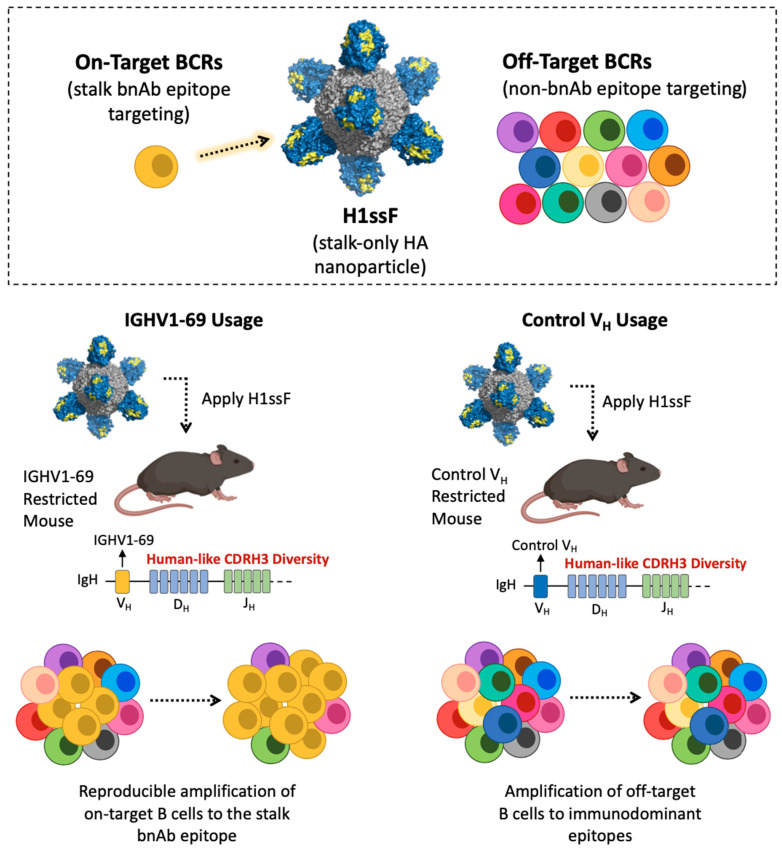
Pathway amplification of on-target B cells. (Top) Key depicting on-target B cells (dark yellow) which target the Group 1 stalk bnAb epitope on H1ssF (yellow). Off-target B cells (various colors) are defined as those that engage non-bnAb epitopes on H1ssF. (Bottom) Schematic depicting pathway amplification of stalk bnAb responses in mice with human-like CDRH3 diversity but constrained to select human V_H_ genes (IGHV1-69 or control V_H_).

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
