# Peer review of "Antibody Focusing to Conserved Sites of Vulnerability: The Immunological Pathways for ‘Universal’ Influenza Vaccines"

_vaccines, 2021, doi:10.3390/vaccines9020125_

Round 1
Reviewer 1 Report
In this review, Sangesland and Lingwood nicely laid out the immunological challenges for universal influenza vaccines (Fig. 2), summarized the biology of potential universal vaccine targets (Fig. 3), and described the rationally designed strategies for addressing the issues facing the developments of universal influenza vaccines (Figs. 4 and 5). Descriptions are concise, and figures are clear and to the points. This very insightful review will be helpful not only for researcher/educators in the field but also for the readers outside the field who are interested in universal influenza vaccines. The manuscript is suitable for vaccines and can be published as it is.
Supplementary comments:
I tried to write a conventional comment, which normally contains comments on strength, weakness, and a list of specific comments. Believe me, I wrote all my past review comments in that way. However, after reading this manuscript, I believed that it was not necessary to try to follow that conventional formula and try to make up some issues that are in fact not issues. The review is clear, insightful, and was perfectly written. If you do not have confidence with my review, please send it to another reviewer to seek additional comments. I am confident that you will receive similar comments as mine.
Author Response
We thank the reviewer for their comments, and hope that others in the field will find the similarly insightful and useful.
Reviewer 2 Report
Maya Sangesland and Daniel Lingwood have comprehended the review entitled “Antibody Focusing to Conserved Sites of Vulnerability: The Immunological Pathways for ‘Universal’ Influenza Vaccines”.
Authors have nicely provided the introduction about flu viruses, and also different vaccine targeting strategies, for the development of universal influenza vaccines.
I endorse the manuscript for the publication.
A few minor checks are needed before publication.
Few sentences, throughout the manuscript, are too wordy. It isn't easy to understand. Please try to reduce as much as possible.
Figure 1b: Influenza B ------ Influenza A
Line 95: delete the repeated word “this”
Line 173-174: Please add a note on the effect of pH on HA stability, which will give more clarity to the readers.
Line 174-180: Please add a balanced sentence with “antibody-dependent enhancement risk”. A single sentence would be ideal.
Author Response
Authors have nicely provided the introduction about flu viruses, and also different vaccine targeting strategies, for the development of universal influenza vaccines.
I endorse the manuscript for the publication.
Response – We greatly appreciate this endorsement.
A few minor checks are needed before publication.
Few sentences, throughout the manuscript, are too wordy. It isn't easy to understand. Please try to reduce as much as possible
Response – We have reduced the length of some sentences (see also responses to Reviewer 3.
Figure 1b: Influenza B ------ Influenza A
Response – “Influenza B” is actually a branch of the tree. We have adjusted the figure to make this clear
Line 95: delete the repeated word “this”
Response – We have made the correction, our thanks to the reviewer for pointing this out.
Line 173-174: Please add a note on the effect of pH on HA stability, which will give more clarity to the readers
Response – We have modified to:
Lines 171-176
“This interface epitope is hidden within the trimeric prefusion conformation but may become transiently accessible through dynamic conformational changes associated with “breathing”, which has been measured for HA trimers at neutral pH [112]. During infection, the virus is internalized within acidic endosomes where the dynamic fluctuations give way to pH-triggered rearrangements in HA2, leading to fusion of the viral and cellular membranes [112].”
Line 174-180: Please add a balanced sentence with “antibody-dependent enhancement risk”. A single sentence would be ideal.
Response – We have added:
Lines 184-186
“Such synergy may also be balanced by antibody-dependent enhancement risks, such as enhanced infection of monocytes and myeloid DCs, as recently reported for anti-HA antibodies from macaques [121].”
Reviewer 3 Report
This is quite a well worked area with coverage in the last couple of years of various attempts to generate a "universal" flu vaccine. The recent Palese paper has also added new direct experimental data. However, what marks this review out is the focus on the immunological pathways that could give rise to amplification of otherwise restricted antibody lineages, very nice supporting figures and a very comprehensive reference list. It's also well written.
My only comment would be that, although almost all focused on influenza, reference to HIV pops up now and again without any explanation. The authors may care to rephrase slightly. For example "Structure-based influenza and HIV
immunogens applied..." might be better as "Structure-based influenza immunogens applied within transgenic mouse systems bearing user-defined B cell receptor (BCR) repertoires, as well as similar previous studies with HIV, have experimentally demonstrated....". It just sets the HIV work as historical so that it does not detract from the influenza focus. But it's not a big point.
Author Response
This is quite a well worked area with coverage in the last couple of years of various attempts to generate a "universal" flu vaccine. The recent Palese paper has also added new direct experimental data. However, what marks this review out is the focus on the immunological pathways that could give rise to amplification of otherwise restricted antibody lineages, very nice supporting figures and a very comprehensive reference list. It's also well written.
My only comment would be that, although almost all focused on influenza, reference to HIV pops up now and again without any explanation. The authors may care to rephrase slightly. For example "Structure-based influenza and HIV
immunogens applied..." might be better as "Structure-based influenza immunogens applied within transgenic mouse systems bearing user-defined B cell receptor (BCR) repertoires, as well as similar previous studies with HIV, have experimentally demonstrated....". It just sets the HIV work as historical so that it does not detract from the influenza focus. But it's not a big point.
Response – we thank the reviewer and have adjusted to:
Lines 105-110
“Structure-based influenza immunogens applied within transgenic mouse systems bearing user-defined B cell receptor (BCR) repertoires, along with similar approaches using HIV antigens, has enabled experimental manipulation of these parameters [84-92]. This work indicates that the immunogenicity of a given epitope is a function of the frequency and affinity of the on-target versus off-target BCRs present in the antigen naïve germline repertoire.”